# Role of PGC-1α in the Mitochondrial NAD^+^ Pool in Metabolic Diseases

**DOI:** 10.3390/ijms22094558

**Published:** 2021-04-27

**Authors:** Jin-Ho Koh, Jong-Yeon Kim

**Affiliations:** Department of Physiology, College of Medicine, Yeungnam University, Daegu 42415, Korea

**Keywords:** mitochondria, PGC-1α, NAD^+^, SIRTs, metabolic disease

## Abstract

Mitochondria play vital roles, including ATP generation, regulation of cellular metabolism, and cell survival. Mitochondria contain the majority of cellular nicotinamide adenine dinucleotide (NAD^+^), which an essential cofactor that regulates metabolic function. A decrease in both mitochondria biogenesis and NAD^+^ is a characteristic of metabolic diseases, and peroxisome proliferator-activated receptor γ coactivator 1-α (PGC-1α) orchestrates mitochondrial biogenesis and is involved in mitochondrial NAD^+^ pool. Here we discuss how PGC-1α is involved in the NAD^+^ synthesis pathway and metabolism, as well as the strategy for increasing the NAD^+^ pool in the metabolic disease state.

## 1. Introduction

Mitochondria are powerhouses that generate the majority of cellular ATP via fatty acid oxidation, tricarboxylic acid (TCA) cycle, electron transport chain (ETC), and ATP synthase. Mitochondrial dysfunction is linked to metabolic diseases and health issues, including insulin resistance and type 2 diabetes, cancer, Alzheimer’s disease, and others [1,2,3].

Nicotinamide adenine dinucleotide (NAD^+^) is an essential cofactor that regulates metabolic function, and it is an electron carrier and signaling molecule involved in response to alterations in the cellular metabolic redox state, including muscle contraction, high-fat diet (HFD), insulin resistance, and type 2 diabetes mellitus (T2DM) [4]. NAD^+^ plays a key role in cellular signaling and regulation of metabolism in glycolysis, oxidative phosphorylation, the TCA cycle, and DNA repair [5,6,7,8]. Moreover, NAD^+^ is reduced to NADH by accepting two electrons and a proton from glycolysis and the TCA cycle, and mitochondrial NADH is oxidized through mitochondrial respiratory complex I (NADH ubiquinone oxidoreductase) in the ETC [9]. This is one of the essential steps during oxidative phosphorylation; therefore, an optimal ratio of NAD^+^/NADH is required for mitochondrial metabolism [10,11,12], and lower NAD^+^ levels and dysregulation of NAD^+^/NADH ratio can be one of the reasons for developing metabolic diseases and T2DM [9]. In particular, NAD^+^ biosynthesis and its function crucially influence the bioenergetic process in mitochondria [13] and are clearly linked to peroxisome proliferator-activated receptor γ coactivator 1-α (PGC-1α) [14,15,16,17,18]. In this review, we focus on how the NAD^+^ pool and PGC-1α regulate mitochondrial health and function in metabolic diseases and T2DM.

## 2. NAD^+^–SIRT1–PGC-1α Pathway in Metabolic Diseases

Plasma glucose homeostasis is critical for the functioning of mammalian organisms; thus, glucose levels should be strictly regulated according to nutrient conditions and energy demands. To maintain glucose homeostasis at the cellular level and to adapt to various challenges such as high-nutrient condition, disuse, and sarcopenia, it is necessary to improve or stabilize mitochondrial function, number, and size, which are important for maintaining the cellular NAD^+^ pool [16,19,20,21,22,23].

PGC-1α is a master regulator that interacts with various transcription factors involved in cellular metabolic functions [24]; thus, PGC-1α mediates the transcriptional activity and biological response related to them [24]. SIRT1 (*Sirtuin 1*) is involved in the regulation of systemic metabolism via the control of glucose and lipid homeostasis by deacetylating various targets, especially PGC-1α [25]. Therefore, the NAD^+^–SIRTs–PGC-1α pathway plays a vital role in cellular metabolic function. NAD^+^ depletion is a characteristic of diabetes [26], and *sirtuins*, including SIRT1-3 and SIRT6, influence cellular functions such as glucose metabolism, mitochondrial function, and oxidative stress [25,27,28,29]. It is well documented that PGC-1α expression is reduced in T2DM muscle [30,31,32].

The NAD^+^ pool is important for cell physiological and metabolic functions for cell integrity; however, metabolic diseases, such as insulin resistance in tissues and diabetes, increase NAD^+^ consumption; thus, lower levels of cellular NAD^+^ are clearly linked to metabolic diseases [33,34,35,36,37,38,39]. In this context, SIRT1, which consumes NAD^+^ for cellular metabolic function, is downregulated in several cells and tissues, including myotubes, HEK293, peripheral blood mononuclear cell, human skeletal muscle, and adipose tissue, in insulin-resistant states [39,40,41]. A previous study has shown that SIRT1 regulates glucose homeostasis by regulating the secretion of insulin and protecting beta (β)-cells in the pancreas [42], enhancing mitochondrial biogenesis and glucose uptake in skeletal muscle [43], and promoting glucose production and fatty acid oxidation in the liver [44]. β-cell-specific SIRT1 overexpression in mice improves insulin secretion and glucose tolerance in response to glucose [42]. Age-related downregulation of SIRT1 activity due to a lack of systemic NAD^+^ biosynthesis results in a decrease in insulin secretion from β-cells in response to glucose; however, treatment with nicotinamide mononucleotide (NMN), which is a derivative of niacin and an intermediate in NAD^+^ biosynthesis in the salvage pathway, restores insulin secretion and improves glucose tolerance in aged mice with β-cell-specific SIRT1 overexpression. Therefore, SIRTs regulate glucose–lipid metabolism and mitochondrial biogenesis via PGC-1α [42,43,44,45]. Overall, NAD^+^ boosting can be one of the strategies to improve metabolic dysfunction via SIRTs–PGC-1α; therefore, we will discuss the role of the SIRTs–PGC-1α pathway in increasing NAD^+^ biosynthesis and decreasing NAD^+^ consumption.

## 3. NAD^+^ Biosynthesis

Cellular NAD^+^ availability is maintained by the regulation of NAD^+^ biosynthesis and degradation. There are five major precursors and intermediates in NAD^+^ synthesis in mammals; tryptophan (Trp), nicotinamide (NAM), nicotinic acid (NA), nicotinamide riboside (NR), and nicotinamide mononucleotide (NMN); they stimulate NAD^+^ synthesis via different pathways [46] (Figure 1). These pathways can synthesize 300–800 µM of cellular NAD^+^, depending upon the tissue and organ [47,48,49,50]. In addition, NAD^+^ can be resynthesized from an intermediate, such as NMN, of NAD^+^. This section focuses on the NAD^+^ biosynthesis pathway.

### 3.1. The Salvage Pathway

The NAD^+^ salvage pathway recycles metabolites produced by cellular catabolism and uses nicotinamide (NAM) [5,51]. The first process involves the addition of a phosphoribosyl moiety to NAM by NAM phosphoribosyltransferase (NAMPT), resulting in NAM mononucleotide (NMN) (Figure 1). The final step of the salvage pathway is processed by NMN adenylyltransferase (NMNAT), which catalyzes NMN into NAD^+^ (Figure 1). NR can also be used as a precursor for NMN by NR kinase (NRK) in the salvage pathway [52]. A few studies have revealed that PGC-1α is closely linked to the NAD^+^ salvage pathway (Figure 1), and metabolites in the NAD^+^ salvage pathway, such as NR and NMN, have been shown to upregulate the SIRT1–PGC-1α pathway in the liver [36] and bone [53]. Moreover, NAMPT is a rate-limiting enzyme responsible for the conversion of NAD^+^ in the salvage pathway [54,55] and is associated with PGC-1α to generate NAD^+^ in the renal epithelium [18] and regulates mitochondrial biogenesis via NAD metabolism in the skeletal muscle [56]. However, further studies are required to understand the relationship between NAMPTs and PGC-1α in the NAD^+^ salvage pathway in various tissues. NMN treatment has also been shown to alleviate osteogenic inhibition and promote the expression of SIRT1 and PGC-1α, whereas these beneficial effects of NMN were reversed when SIRT1 and PGC-1α were decreased [53]. Thus, the SIRT1–PGC-1α pathway may be involved in the conversion of NMN into NAD^+^; however, the mechanism by which SIRT1–PGC-1α regulates NMNAT to convert NMN into NAD^+^ is still poorly understood.

### 3.2. The Preiss–Handler Pathway

NA is also one of the precursors for NAD^+^ synthesis, and this NAD^+^ synthesis pathway is called the Preiss–Handler pathway, which consists of three steps [54]. In the first step, NA is converted into NA mononucleotide (NAMN) by NA phosphoribosyltransferase (NAPRT), followed by the conversion of NAMN to NA adenine dinucleotide (NAAD) by NMNATs (Figure 1). Both NA and NAM can be converted to NAD^+^ by NMNAT in the salvage or Preiss–Handler pathway. In the final step, NAAD is converted to NAD^+^ by NAD synthetase (NADS) [9] (Figure 1). There are few studies on the relationship between PGC-1α and the Preiss–Handler pathway; however, NMNATs are also involved in this pathway to catalyze NAMN in NAAD [57]. Since PGC-1α possibly influences NMNATs that partially catalyze NMN to NAD^+^ in the salvage pathway [53,58], PGC-1α is likely to be involved in the functioning of NMNATs in the Preiss–Handler pathway as well.

### 3.3. The De Novo Synthesis (DNS) Pathway

Trp, an essential amino acid, is a source of NAD^+^. Trp is converted into N-formylkynurenine (NFK) by indoleamine-2,3-dioxygenase (IDO) or tryptophan-2,3-dioxygenase (TDO) (Figure 1). NFK is subsequently converted to 2-amino-3-carboxymuconate-6-semialdehyde (ACMS) by four consecutive enzymatic steps (Figure 1). ACMS is then transformed into quinolinic acid (QA) by a nonenzymatic reaction following which quinolinate phosphoribosyl transferase (QPRT) catalyzes the conversion of QA to NAMN; this is then converted to NAAD followed by the synthesis of NAD^+^ as described in the Preiss–Handler pathway [54] (Figure 1). Thus, the DNS pathway consists of seven steps followed by the step involved in the Preiss–Handler pathway. A previous study has shown that the DNS of NAD^+^ enhances mitochondrial function and improves health [59]. Previous studies have also shown that PGC-1α coordinately increases the enzymes for NAD^+^ DNS from amino acids, whereas the lack of PGC-1α downregulates the DNS pathway in the kidney [18]. The DNS of NAD^+^ proceeds from the degradation of tryptophan to kynurenine, and a previous study has shown that PGC-1α1, a PGC-1α isoform, enhanced peripheral kynurenine catabolism [60] (Figure 1). Thus, increased kynurenine, induced by PGC-1α1, can be converted into kynurenic acid or 3-hydroxykynurenine to synthesize NAD^+^ (Figure 1). Therefore, we suggest that PGC-1α1 is likely to trigger DNS through enhanced kynurenine catabolism; however, it is necessary to find direct evidence to understand that PGC-1α1 is involved in NAD^+^ de novo synthesis.

### 3.4. NAD^+^ Consumption as a Cosubstrate

Intake of dietary tryptophan or less than 20 mg of niacin can meet the daily requirements for NAD^+^ biosynthesis [61].

NAD^+^ is a cosubstrate for the SIRT family of deacylases; this action effectively improves mitochondrial function, metabolism, and aging with various health benefits. NAD^+^ is used as a cosubstrate when the SIRT family generates the deacylated substrate NAM by removing an acyl group from its substrate [9]; thus, the SIRT family consumes cellular NAD^+^ (Figure 1). Besides *sirtuins*, cyclic ADP-ribose (cADPR) synthases also consume NAD^+^ (Figure 1). cADPR uses NAD^+^ to generate cADPR, which is a cellular regulator of Ca^2+^ homeostasis [62]. CD38 and its homolog CD157 are the most well-known cADPR synthases, which indirectly influence various Ca^2+^-dependent activities for muscle contraction, immune responses, cell proliferation, and insulin secretion from pancreatic β-cells [63] (Figure 1); thus, CD38-deficient mice exhibit approximately 30-fold higher cellular NAD^+^ levels than wild-type [64]. Other major NAD^+^-consuming enzymes belong to the poly(ADP-ribose) polymerase (PARP) protein family, which plays an essential role in DNA repair and preservation of genomic integrity [65] (Figure 1). PARPs regulate poly ADP-ribosylation (PARylation), which is a fully reversible post-translational modification that plays a key role in cellular physiology [66]. In this process, a large ADP-ribose polymer is added to the target protein, and NAD^+^ is concomitantly hydrolyzed into NAM [9].

Previous studies have shown that SIRT1 binds to promoters of genes that are regulated by NAMPT, NMNAT-1, and SIRT1 and that the histone deacetylase activity of SIRT1 is controlled by NAMPT and NMNAT-1 at these promoters [66], indicating that SIRT1 plays a key role in mediating gene expression regulated by NAMPT and NMNAT-1. Furthermore, the roles of these factors are complementarily regulated by each other. Interestingly, PARP1 also regulates the binding of NMNAT1 to the target gene [67], suggesting that NAD^+^ production and consumption are regulated by competition between SIRT1 and PARP1 during the NAD^+^ salvage pathway or NAD^+^ consumption pathways, such as transcriptional regulation or DNA repair. SIRT1 consumes NAD^+^ during transcription and cooperates with PGC-1α to regulate mitochondrial biogenesis. PARP-1 inhibition protects the diabetic heart via SIRT1/PGC-1α [38]. Thus, we speculate that PGC-1α regulates not only NAD^+^ consumption via SIRT1 but also PARP and NAD^+^ production via NAMPT and NAMPT. However, further studies are required to understand the role of PGC-1α in NAD^+^ production and consumption.

Glycolysis also plays a significant role in the transfer of NAD^+^ to the mitochondria and the maintenance of the mitochondrial NAD^+^ pool. During glycolysis, two NAD molecules are necessary for glyceraldehyde-3-phosphate dehydrogenase (GAPDH) to oxidize glyceraldehyde-3-phosphate to 1,3-bisphosphoglycerate [6]. Two NADH molecules produced in the cytosol by glycolysis are delivered into the mitochondrial matrix to provide reducing equivalents for the TCA cycle and ETC [13,48,68,69]. Unlike NAD^+^, NADH can freely enter the intermembrane space either via the glycerol-3-phosphate shuttle of the inner mitochondrial membrane, the malate–aspartate shuttle, or by itself [6,70,71]. NAD^+^ is reduced to NADH in the TCA cycle, followed by NADH oxidation to NAD^+^ by NADH-ubiquinone oxidoreductase in the ETC. Thus, the mitochondrial NAD^+^ pool is important for maintaining oxidative phosphorylation and ATP levels. In addition, NAD^+^ transfer into the mitochondria from the cytosol is one of the critical ways to maintain the NAD^+^ pool in mitochondria and is required for cellular metabolism and cell survival.

## 4. PGC-1α1 Regulates the Mitochondrial NAD^+^ Pool via Malate–Aspartate Shuttle

NAD^+^ levels in mitochondria are important for maintaining metabolic functions and cell survival in oxidative metabolic tissues, including the skeletal muscle, heart, and liver. In metabolic tissues that dominantly use oxidative phosphorylation to generate ATP, NAD^+^ levels in mitochondria should be maintained at a higher level than that in the cytoplasm [13,72,73,74]. Malate dehydrogenase (MDH) and aspartate aminotransferase (AST) form the malate–aspartate shuttle (MAS), which plays a vital role in the exchange of cytosolic NADH for mitochondrial NAD^+^, which is an irreversible step in the exchange of mitochondrial aspartate and cytosolic glutamate and a proton by the aspartate–glutamate carrier (AGC) [75,76,77] (Figure 2). Thus, MAS can regulate mitochondrial NAD^+^ pool, and since the mitochondrial NAD^+^ pool is well maintained and higher than that in the cytoplasm (cytosolic/nuclear NAD^+^ levels are ~100 µM, while mitochondrial NAD^+^ levels are ~250 µM) [74], mitochondrial numbers and size regulate NAD^+^ levels in metabolic tissue. Mitochondria function as an NAD^+^ warehouse. Even if a large amount of NAD^+^ is depleted from the cytoplasm, mitochondrial NAD^+^ levels can be conserved for at least 24 h and possibly up to three days [47,48,72,78,79,80], indicating that mitochondrial NAD^+^ has specific roles in metabolism and is separated from the cytoplasm.

This speculation leads to the hypothesis that MAS-induced increase in the mitochondrial NAD^+^ pool can increase the expression of *sirtuin* family (SIRT1-7) members and PGC-1α. Indeed, a previous study has shown that MAS regulates the intracellular NAD^+^/NADH ratio; moreover, calorie restriction increases the mitochondrial NAD^+^ pool and SIRT2 expression via MAS [75].

An AGC1 knockout study has been shown to decrease the cellular NAD^+^/NADH ratio and impair aspartate delivery to the cytosol [81]. Moreover, muscle-specific PGC-1α1 overexpression in mice enhances the expression of SLC25A12, another gene name for AGC1 [71]. MAS activated by PGC-1α1 rigidly maintains the NAD^+^ pool to maintain oxidative metabolism when energy demand is high (Figure 2), such as muscle contraction during exercise. Trained muscle has a higher level of NAD^+^ pool [20], and PGC-1α1 regulates mitochondrial biogenesis and cellular oxidative metabolism. Overall, these studies lead us to speculate that MAS is linked to mitochondrial and metabolic functions. Furthermore, a previous study has shown that deficiency in MDH, which plays an essential role in the MAS and TCA cycle, is a metabolic defect characterized by a severe neurodevelopmental phenotype [82].

## 5. Role of PGC-1α in NAD^+^ Metabolism in Metabolic Diseases

SIRT1 is a nicotinamide adenosine dinucleotide (NAD)-dependent deacetylase that removes acetyl groups from histone and nonhistone proteins [83]. It potentially mediates the effects of calorie restriction on health benefits for longevity [84], and exercise also increases NAD^+^/NADH turnover [85]. Increased NAD^+^/NADH turnover rate by glycolysis, TCA cycle, and mitochondrial oxidative phosphorylation system during cellular high energy-demand and increased NAD^+^ by SIRT1 increase mitochondrial biogenesis [4], which induces the salvage pathway to regenerate NAD^+^, because this pathway can quickly regenerate NAD^+^ from NAM in two steps. NAMPT and NMNAT are enzymes involved in the salvage pathway.

The findings to date show that PGC-1α has the most influence on the salvage mechanism of NAD^+^ metabolism. PGC-1α is activated by specific SIRTs, and PGC-1α promotes NAD^+^ re-biosynthesis via the salvage pathway and increases mitochondrial biogenesis, thereby improving mitochondrial function and protecting against high fat diet induced obesity [86]. Here, we focus on the PGC-1α mechanism in various NAD^+^ consumption and biosynthesis pathways related to metabolic diseases.

### 5.1. NAMPT–PGC-1α

NAMPT protein in human skeletal muscle is positively correlated with whole-body insulin sensitivity and negatively correlated with body fat [87]. NAMPT and NAD^+^ levels were decreased by HFD feeding in the liver and white adipose tissue (WAT) [33], whereas fasting and calorie restriction (CR) increased NAMPT expression in the liver and skeletal muscle [47,88,89,90,91]. NAMPT and NMN have been shown to increase insulin secretion in human islets [92], and NAD^+^ levels that are mediated by NAMPT were decreased in T2DM mice, whereas administration of NMN, which is converted into NAD^+^ by NMNATs, improved glucose tolerance [33].

We suggested the hypothesis that NAMPT and NMNAT in the salvage pathway are related to PGC-1α in the production of NAD^+^ (Figure 1). NAMPT is enhanced by exercise or calorie restriction in an AMPK-activation-dependent manner [87,93]. AMPK is an energy sensor that regulates PGC-1α transcription [94] and SIRT1 activity [93], and NAMPT expression and SIRT1 activity were increased during glucose restriction in mouse myoblasts in an AMPK-dependent manner [91], suggesting that NAMPT could be involved in mitochondrial biogenesis via PGC-1α. This study showed that PGC-1α is likely to mediate NAMPT expression to regulate the salvage pathway for NAD^+^ biosynthesis; however, it does not provide direct evidence that PGC-1α regulates NAMPT expression and is involved in the NAMPT activity to control the salvage pathway; thus, further studies are required to clarify this.

### 5.2. PARP1–PGC-1α

PARP1 activation consumes NAD^+^ for DNA repair. Although the SIRT family and PARPs compete for NAD^+^, PARPs have more advantages than SIRT in using the limited NAD^+^ resources, because the Michaelis constant (K_m_) values of PARPs for NAD^+^ are lower than the physiological range of NAD^+^, which is lower than SIRT1 [9,95]. Indeed, previous studies have shown that NAD^+^ availability limits SIRT activity when PARPs are overactivated during aging or after DNA damage [21,96,97,98]. Therefore, the process of repairing DNA damage takes precedence over the specific roles of the SIRT family. However, in case of sufficient NAD^+^ levels, SIRT family and PGC-1α could block PARP activation, and PARP1 inhibition has been shown to protect the diabetic heart via activation of SIRT1–PGC-1α [38,99], indicating that the SIRT1–PGC-1α pathway could inhibit cellular NAD^+^ consumption due to PARP activation when cellular NAD^+^ levels are sufficient. In other cases of enhanced NAD^+^/NADH turnover rate, including exercise and calorie restriction, PGC-1α is elevated in cells and inhibits cellular oxidative damage and protects the DNA against various challenges such as reactive oxygen species (ROS) or metabolic and physiological dysfunction [14]; thus, we suggest that PGC-1α could facilitate a cellular condition that prevents unnecessary NAD^+^ consumption by PARP and saves cellular NAD^+^ level (Figure 1).

In addition, deficiency of PARP1 increased mitochondrial biogenesis and energy expenditure and therefore protected mice against HFD-induced metabolic disease [11]. PARP1 inhibition protected mice against streptozotocin (STZ)-induced β-cell dysfunction by maintaining NAD^+^ levels and glucose tolerance [100]. Similarly, treatment with PARP1 and PARP2 inhibitors has been shown to increase muscle mitochondrial function and exercise capacity [101,102].

### 5.3. CD38–PGC-1α

CD38 is one of the main NAD^+^-consuming enzymes in mammalian tissue [9]. Previous studies have shown that CD38-knockout mice exhibit higher NAD^+^ levels and reduced metabolic syndrome in obesity [103], and CD38 inhibitors elevate cellular NAD^+^ levels and improve various aspects of both glucose and lipid homeostasis [104]. In addition, CD38 plays an essential role in aging-induced NAD^+^ decline in mice via a SIRT3-dependent mechanism [105], and SIRT3 increases mitochondrial biogenesis and decreases ROS in a PGC-1α-dependent manner [106]. Since SIRT3 alone cannot increase mitochondrial biogenesis and influence ROS production, it appears that SIRT3 cooperates with PGC-1α to regulate CD38. However, this relationship is poorly understood, and further studies are required (Figure 1).

### 5.4. ACMSD

In de novo synthesis of NAD^+^, α-amino-β-carboxymuconate-ε-semialdehyde (ACMS) is converted into quinolinic acid (QA) via a nonenzymatic reaction (spontaneous cyclization), and it is also decarboxylated via ACMSD [9]. Thus, enhanced ACMSD can reduce the rate of conversion of ACMS into QA to produce NAD^+^ (Figure 1). It is well known that inhibition of ACMSD protects hepatocytes against apoptosis induced by high doses of fatty acids in a SIRT1-dependent mechanism, and ACMSD inhibition also increases mitochondrial gene expression in the liver [59]. Previous studies have shown that PGC-1α drives de novo synthesis of NAD^+^ for renal protection [18]. Thus, PGC-1α appears to be involved in ACMSD activity in numerous ways (Figure 1), and further studies are required to understand this.

### 5.5. NNMT

NAM N-methyltransferase (NNMT), which is a *sirtuins* and NAD^+^ inhibitor, regulates NAD^+^ biosynthesis by catalyzing NAM into MNA. NAM is converted into NMN by NAMPT in the salvage pathway; however, NNMT promotes the conversion of NAM into 1-methyl-nicotinamide (MNA), which cannot be used for NAD^+^ synthesis [9]. Indeed, NNMT knockdown increases NAD^+^ levels via enhanced NAMPT in adipose tissue [107]. Constant NNMT activation decreases NAD^+^ content, SIRT3 activity, and PGC-1α expression in the liver [34].

NNMT expression has been shown to be negatively correlated with glucose transporter 4 (GLUT4) in adipose tissue [107], and NNMT expression was increased in ob/ob, db/db, and HFD-mice when compared to lean mice, whereas NNMT knockdown in white adipose tissue and liver protected against HFD-induced obesity via enhanced SIRT1 target gene expression and energy expenditure [107]. Thus, SIRTs–PGC-1α may influence NNMT function.

### 5.6. NADH–NQO1–PGC-1α Pathway and NAD+ Level

NADH quinone oxidoreductase 1 (NQO1) dysfunction is linked to metabolic dysfunction [108,109,110,111]. NQO1 binds to 20S proteosome, which inhibits protein degradation [112,113]. NQO1 determines PGC-1α basal levels in muscles and PGC-1α-induced levels in liver cells [17]. Moreover, NADH binds to NQO1 and protects PGC-1α degradation; thus, NQO1 regulates PGC-1α expression and activity via emerging redox signaling [16,17] (Figure 3). Moreover, NQO1 catalyzes the transformation of quinones into hydroquinones via NADH as an electron donor, resulting in increased cellular NAD^+^ levels. A previous study has shown that a lack of NQO1 decreases NAD^+^ content in the liver and kidneys and increases insulin resistance [109], whereas enhanced NQO1 activation has been shown to increase NAD^+^ levels in cells [114,115] and protect obese mice via enhanced glucose and lipid metabolism [116]. These results suggest that enhanced NQO1–PGC-1α mediates the maintenance of cellular NAD^+^ levels and protects against metabolic dysfunction.

## 6. NAD^+^–SIRT1–PGC-1α Pathway in Diabetes

Mitochondrial function is involved in whole-body and cellular glucose homeostasis, and mitochondrial functions are decreased in states of insulin resistance and diabetes [2]. It is well established that an increase in mitochondrial function by exercise training in various metabolic tissue, including skeletal muscle and adipose tissue, of the insulin resistance or diabetes subjects improves glucose homeostasis [117,118]. Thus, mitochondrial function is important for glucose homeostasis.

SIRTs directly interact with PGC-1α and deacetylate at specific lysine residues in an NAD^+^-dependent manner, activating PGC-1α [119]. Thus, NAD^+^–SIRTs–PGC-1α pathway plays a critical role in glucose homeostasis in diabetes and lifespan [119]. As we discussed above, since mitochondria function as a warehouse of NAD^+^, and PGC-1α regulates mitochondrial biogenesis in metabolic tissue, NAD^+^, SIRT1, and PGC-1α play an important role in their function of regulating glucose homeostasis in each other. In metabolic tissue, including skeletal muscle, liver, and adipose tissue, of T2DM state, PGC-1α protein and NAD^+^ levels are downregulated, as is SIRT1 activity [33,120,121,122]. These shifts are clearly linked to the mitochondrial functions, numbers, and size [33,120,121,122].

As the strategy to increase cellular NAD^+^ levels, the biosynthesis of NAD^+^ should be increased, and its consumption should be prevented. Daily intake of NAD^+^ precursor is one of the strategies to increase NAD^+^ synthesis and improve metabolic function in diabetes. Indeed, a previous study has shown that tryptophan supplementation increases lifespan through NAD^+^ de novo synthesis [59], and acipimox, an NAD^+^ precursor, can directly improve skeletal muscle mitochondrial function in T2DM patients [123]. Mice receiving NR were protected against high fat diet induced weight gain and had higher insulin sensitivity with increased mitochondrial content in skeletal muscle and brown adipose tissue [124]. Niacin supplementation can improve lipid profiles in T2DM patients [125]. These data suggest that supplementation of NAD^+^ precursor can be one of the strategies to improve insulin resistance and treat T2DM (Figure 4).

It is widely accepted that exercise can provide many health-beneficial effects for T2DM patients; in this context, exercise could increase the cellular NAD^+^ pool in an indirect way via endogenous enzyme alteration to activate its synthesis or protect the NAD^+^ pool against overconsumption. Indeed, a previous study has revealed that trained muscle has a higher level of NAD^+^ pool, and aerobic and resistance exercise training improves the capacity of NAD^+^ salvage pathway in aged human skeletal muscle [20]. AMPK is activated by exercise [126], and it is well evidenced that AMPK regulates the gene expression related to energy metabolism in mouse skeletal muscle via coordination with SIRT1 and enhances its activity by increasing cellular NAD^+^ pool [93]. Exercise increases NAMPT in human skeletal muscle [87], and NAMPT overexpression in mice skeletal muscle increases muscle NAD^+^ levels and protects against body weight gain in high-fat-fed mice, as well as increasing mitochondrial gene expression and endurance capacity [127]. Exercise-training-induced increase in mitochondrial biogenesis in tissue also protects against overconsumption of NAD^+^ because the mitochondrial capacity for NAD^+^ conservation is higher than that of the cytoplasm [47,48,72,78,79,80] (Figure 4).

## 7. Conclusions

NAD^+^ boosting can be a strategy to improve mitochondrial and metabolic function and to protect against various metabolic diseases, such as insulin resistance and T2DM, via SIRTs–PGC-1α. NAD^+^ biosynthesis should be increased and NAD^+^ consumption should be decreased to increase cellular NAD^+^ levels. In addition, exercise training can increase cellular NAD^+^ levels. However, high-intensity exercise seems to require NAD^+^ supplements because it requires more NAD^+^ to produce more energy, for increased gene expression during exercise, and to repair damaged muscle cells after high-intensity exercise. In particular, after high-intensity exercise, NAD^+^ is required for cell recovery and the repair of damaged muscle cells via CD38 and PARP1, which could inhibit the activity of SIRTs. Sufficient NAD^+^ levels could activate SIRTs, even though CD38 or PARP is activated. Thus, the exercise performance via PGC-1α is beneficial and ensures significant recovery at the cellular level. Therefore, we recommend NAD^+^ supplements to increase the beneficial effects of exercise, especially for aging people and patients with metabolic diseases. Therefore, NAD^+^–SIRTs–PGC-1α may be a therapeutic target for metabolic diseases.

## Figures and Tables

**Figure 1 ijms-22-04558-f001:**
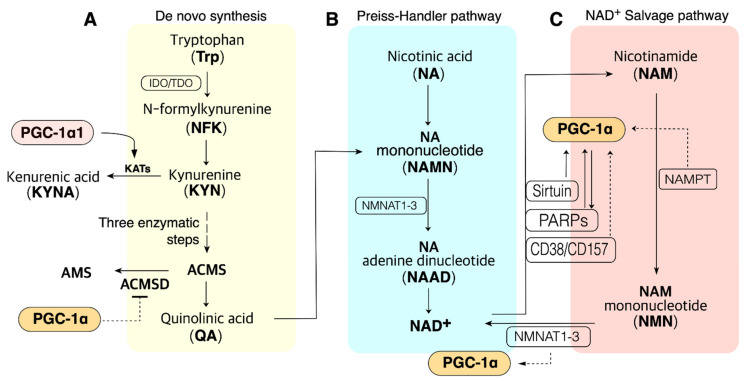
PGC-1α is involved in the pathway of NAD^+^ biosynthesis and consumption in the metabolic tissue such as muscle, liver, and adipose tissue. (**A**). NAD^+^ de novo synthesis from Trp. PGC-1α may promote QA synthesis via blocking ACMSD. PGC-1α1 may promote de novo synthesis via KAT activation. (**B**). NAD^+^ biosynthesis from NA in the Preiss–Handler pathway; QA from de novo synthesis also can be synthesized to NAD^+^ in the Preiss–Handler pathway. (**C**). PGC-1α may influence many enzymes in the salvage pathway. ACMS, α-amino-β-carboxymuconate-ε-semialdehyde; ACMSD, ACMS decarboxylase; NAMPT, nicotinamide phosphoribosyltransferases (three isoforms exist); NMNAT, NMN adenylyltransferase; PARP, poly (ADP-ribose) polymerase. Dashed lines indicate additional evidence is required to reveal the mechanisms.

**Figure 2 ijms-22-04558-f002:**
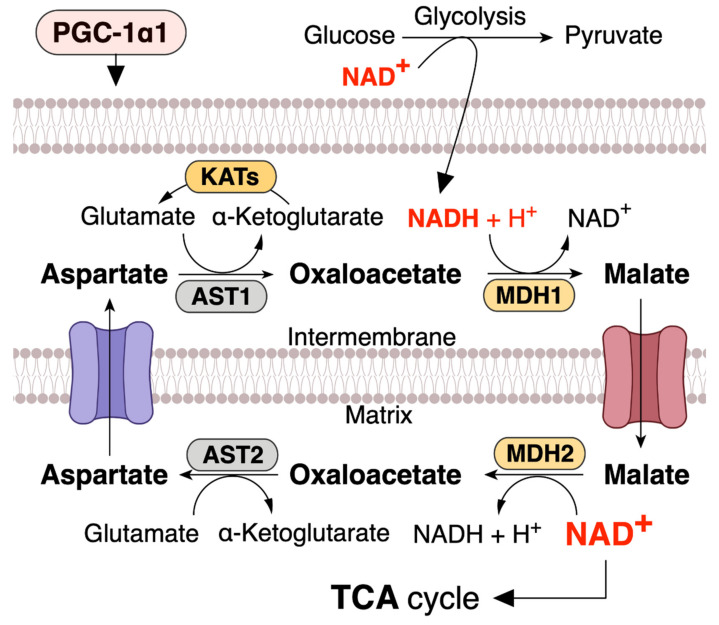
PGC-1α1 regulates malate and aspartate system (MAS) and increases mitochondrial NAD^+^ levels in skeletal muscle. NADH produced during glycolysis enters the mitochondrial matrix via the MAS. MDH, malate dehydrogenase; AST, aspartate aminotransferase; KATs, kynurenine aminotransferases. This concept was referenced from the study by Agudelo et al. (2019) [71].

**Figure 3 ijms-22-04558-f003:**
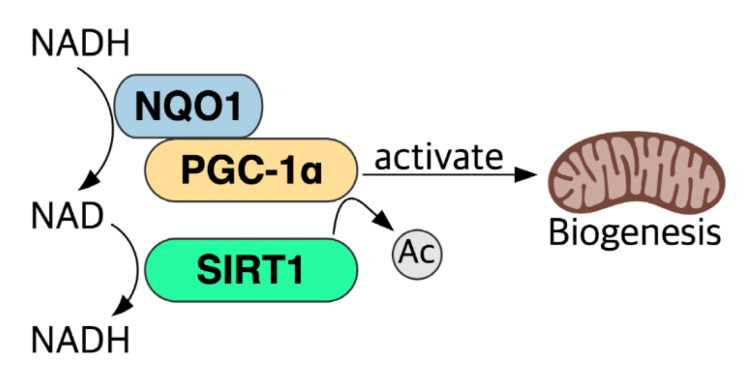
NQO1 cooperates with SIRT1 to activate PGC-1α. NQO1 binds to PGC-1α in an NADH-dependent manner, and SIRT1 activates PGC-1α via the removal of an acetyl group from PGC-1α using NAD^+^.

**Figure 4 ijms-22-04558-f004:**
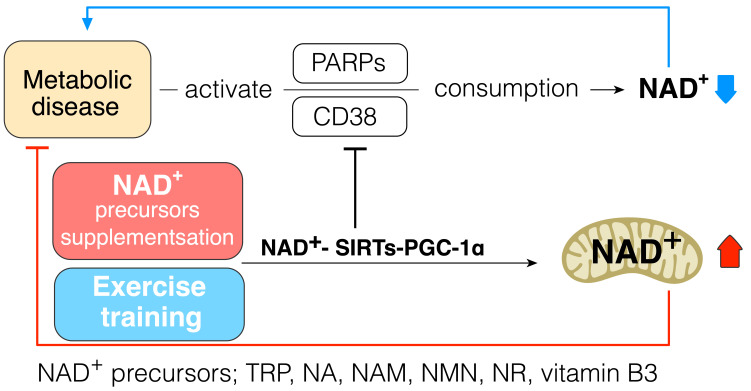
NAD^+^–SIRTs–PGC-1α pathway increases NAD^+^ pool and inhibits NAD^+^ consumption. NAD^+^ precursors: tryptophan (Trp), nicotinic acid (NA), nicotinamide (NAM), NAM mononucleotide (NMN), nicotinamide riboside (NR), vitamin B3; metabolic diseases: insulin resistance, type 2 diabetes.

## Data Availability

Not applicable.

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
