# Peer review of "Role of PGC-1α in the Mitochondrial NAD+ Pool in Metabolic Diseases"

_ijms, 2021, doi:10.3390/ijms22094558_

Round 1
Reviewer 1 Report
In the review titled “Role of PGC-1α in the mitochondrial NAD+ pool in metabolic 2 diseases”, the authors describe how PGC-1α is involved in NAD+ metabolism. Overall, the review is written well, however the authors should try to make the point precise. The sentences are long and sometimes difficult to understand. Please address the following concerns.
As previously mentioned, the NAD+ pool is important for cell physiological and metabolic functions for cell integrity; however, metabolic diseases, such as insulin resistance in tissues and diabetes, increase NAD+ consumption; thus, lower levels of cellular NAD+ are clearly linked to metabolic diseases
Please provide a reference to the following sentence.
In this context, SIRT1, which consumes NAD+ for cellular metabolic function, is downregulated in several cells and tissues in insulin-re-sistant states [29-31].
The authors must provide the details of the tissues and cells.
Also, the authors must change the word “consumes” and replace it with a suitable alternative
To maintain glucose homeostasis at the cellular level and to adapt to various challenges such as high-nutrient condition, disuse, and sarcopenia, it is necessary to improve or stabilize mitochondrial function, number, and size, which are important for maintaining the cellular NAD+ pool.
Please provide a reference to the sentence stated.
A decrease in mitochondrial number and size is likely to cause a lower cellular NAD+ pool and induce metabolic diseases.
Please provide a reference to the sentence stated.
In addition, NAD+ can be resynthesized from an intermediate of NAD+
Please provide the name of the intermediate.
A few studies have revealed that PGC-1α is closely linkedlinks to the NAD+ salvage path- 98 way (Fig. 1), and metabolites in the NAD+ salvage pathway, such as NR and NMN, have 99 been shown to upregulate the Sirt1-PGC-1α pathway in the liver [40] and bone [41].
Please correct for the word “linkedlinks”.
In the Figure1, the authors wrote “PGC-1α is involved in the pathway of NAD+ biosynthesis and consumption”.
Please change the word “consumption” to degradation or other suitable word.
The authors mention three major pathways in Figure 1. They write NMNAT1-3. What does 1-3 designate?
The authors should mention the tissues in the figure where the pathways are predominantly present.
In the Figure 2 legends, the acronym should be written in parentheses following the full form of the name of enzyme.
Thus, the SIRT1-PGC-1α pathway may be involved in the conversion of NMN into NAD+ ; however, the mechanism by which SIRT1-PGC-1α regulates NMNAT to convert NMN into NAD+ is still poorly understood,
The authors might consider mentioning a mechanism they think might operate
There are few studies on the relationship between PGC-1α and the Preiss Handler pathway; however, NMNATs are also involved in this pathway to catalase NAMN in NAAD [45].
Please correct the word “catalase”,
The consumption of less than 20 mg of niacin can meet the daily requirements for 146 NAD+ biosynthesis [48]
The authors state that consumption of niacin can meet the requirements of NAD+.
Similarly, tryptophan consumption should also meet the requirements of NAD+. This should be mentioned
Thus, PGC-1α seems to regulate NAD+ consumption via SIRT1 and PARP and NAD+ production via NAMPT and NAMPT
Please rephrase the sentence.
Thus, the mitochondrial NAD+ pool is important for maintaining oxidative phosphorylation and ATP levels, and NAD+ transfer into the mitochondria from the cytosol results in the NAD+ pool in mitochondria and is important for cellular metabolism and cell survival.
Rephrase the sentence.
NAMPT protein in human skeletal muscle is positively correlated with whole-body insulin sensitivity and negatively correlated with body fat [77], and NAMPT and NAD+ levels were decreased in the liver and white adipose tissue (WAT) by HFD feeding [80], whereas fasting and calorie restriction (CR) increased NAMPT expression in the liver and skeletal muscle [35,79,81-83].
Please rephrase the sentence. Please break it down to two /three sentences.
Author Response
We thank the reviewer for their comments and accordingly revised the manuscript to address these concerns.
I have explained point by point the details of the revisions in the manuscript and my responses to your comments. R; reviewer comment, A; answer.
R1. Please provide a reference to the following sentence.
As previously mentioned, the NAD+ pool is important for cell physiological and metabolic functions for cell integrity; however, metabolic diseases, such as insulin resistance in tissues and diabetes, increase NAD+ consumption; thus, lower levels of cellular NAD+ are clearly linked to metabolic diseases
A1. We are sorry this sentence was removed because duplicated.
R2. The authors must provide the details of the tissues and cells.
In this context, SIRT1, which consumes NAD+ for cellular metabolic function, is downregulated in several cells and tissues in insulin-resistant states [29-31].
A2. We provided the information of tissues and cells as below sentence,
“in several cells and tissues including myotubes, HEK293, peripheral blood mononuclear cell, human skeletal muscle, and adipose tissue in insulin-resistant states [39-41].”
R3. Also, the authors must change the word “consumes” and replace it with a suitable alternative
A3. We thank the reviewer's comment, however, SIRT1 uses the NAD+ for its own function, thus we used the word that “consumes”, which fits our meaning. I will appreciate it if you understand our decision.
R4. Please provide a reference to the sentence stated.
To maintain glucose homeostasis at the cellular level and to adapt to various challenges such as high-nutrient condition, disuse, and sarcopenia, it is necessary to improve or stabilize mitochondrial function, number, and size, which are important for maintaining the cellular NAD+ pool.
A4. As the reviewer commented, we provided the references as below sentence,
“……function, number, and size, which are important for maintaining the cellular NAD+ pool [16,19-23].”
R5. Please provide a reference to the sentence stated.
A decrease in mitochondrial number and size is likely to cause a lower cellular NAD+ pool and induce metabolic diseases.
A5. Since we discussed the sentence in section 4, we removed the sentence.
R6. Please provide the name of the intermediate.
In addition, NAD+ can be resynthesized from an intermediate of NAD+
A6. We provided the name as this sentence; “from an intermediate, such as NMN, of NAD+”
R7. Please correct for the word “linkedlinks”.
A7. We revised the typo as a correct word; “linked”.
R8. Please change the word “consumption” to degradation or other suitable word.
In the Figure1, the authors wrote “PGC-1α is involved in the pathway of NAD+ biosynthesis and consumption”.
A8. We thank the reviewer's comment, however, SIRT1 uses the NAD+ for its own function, thus we used the word that “consumes”, which fits our meaning. I will appreciate it if you understand our decision.
R9. The authors mention three major pathways in Figure 1. They write NMNAT1-3. What does 1-3 designate?
A9. We described that NMNAT1-3 in the figure legend; nicotinamide phosphoribosyltransferases (three isoforms exist).
R9. The authors should mention the tissues in the figure where the pathways are predominantly present.
A9. We revised the legend title of figure 1; “PGC-1α is involved in the pathway of NAD+ biosynthesis and consumption in the metabolic tissue such as muscle, liver, and adipose tissue.” We also revised the legend title of figure 2; "PGC-1α1 regulates malate and aspartate system (MAS) and increases mitochondrial NAD+ levels in skeletal muscle."
R10. In the Figure 2 legends, the acronym should be written in parentheses following the full form of the name of enzyme.
A10. We revised the legend title of figure 2; “PGC-1α1 regulates malate and aspartate system (MAS) and increases mitochondrial NAD+ levels.”
R11. The authors might consider mentioning a mechanism they think might operate
Thus, the SIRT1-PGC-1α pathway may be involved in the conversion of NMN into NAD+ ; however, the mechanism by which SIRT1-PGC-1α regulates NMNAT to convert NMN into NAD+ is still poorly understood,
A11. We thank the reviewer comment, however, we don't have the mechanism to suggest.
R12. Please correct the word “catalase”
There are few studies on the relationship between PGC-1α and the Preiss Handler pathway; however, NMNATs are also involved in this pathway to catalse NAMN in NAAD [45].
A12. We revised the word “catalase”; “catalyze”.
R13. The authors state that consumption of niacin can meet the requirements of NAD+. Similarly, tryptophan consumption should also meet the requirements of NAD+. This should be mentioned
The consumption of less than 20 mg of niacin can meet the daily requirements for 146 NAD+ biosynthesis [48].
A13. We revised the sentence as below sentence;
“Intake of dietary tryptophan or less than 20 mg of niacin can meet…...”
R14. Please rephrase the sentence.
Thus, PGC-1α seems to regulate NAD+ consumption via SIRT1 and PARP and NAD+ production via NAMPT and NAMPT
A14. We rephrase the sentence as below;
“Thus, PGC-1α seems to regulate not only NAD+ consumption via SIRT1 but also PARP and NAD+production via NAMPT and NAMPT.”
R15. Rephrase the sentence.
Thus, the mitochondrial NAD+ pool is important for maintaining oxidative phosphorylation and ATP levels, and NAD+ transfer into the mitochondria from the cytosol results in the NAD+ pool in mitochondria and is important for cellular metabolism and cell survival.
A15. We rephrase the sentence as below;
“Thus, the mitochondrial NAD+ pool is important for maintaining oxidative phosphorylation and ATP levels. In addition, NAD+ transfer into the mitochondria from the cytosol is one of the critical ways to maintain the NAD+ pool in mitochondria and is required for cellular metabolism and cell survival.”
A16. Please rephrase the sentence. Please break it down to two /three sentences
NAMPT protein in human skeletal muscle is positively correlated with whole-body insulin sensitivity and negatively correlated with body fat [77], and NAMPT and NAD+ levels were decreased in the liver and white adipose tissue (WAT) by HFD feeding [80], whereas fasting and calorie restriction (CR) increased NAMPT expression in the liver and skeletal muscle [35,79,81-83].
A16. As the reviewer suggested, sentences were broken down into two, and rephrase as below;
“NAMPT protein in human skeletal muscle is positively correlated with whole-body insulin sensitivity and negatively correlated with body fat [86]. NAMPT and NAD+ levels were decreased by HFD feeding in the liver and white adipose tissue (WAT) [34], whereas fasting and calorie restriction (CR) increased NAMPT expression in the liver and skeletal muscle [46,88-91].”
Reviewer 2 Report
In this manuscript, the authors review the experimental evidence connecting PGC-1alpha and sirtuins (SIRTs) to the regulation of NAD+ under normal and pathological conditions, and based on these data, they make a case for the importance of NAD+/SIRT/PGC-1alpha in metabolic disease and diabetes. Overall, the manuscript is well organized, clearly written and the figures are appropriate and complement the text. There are, however, a number of concerns that limit enthusiasm for the paper in its current form.
- In many instances throughout the text the authors reference other Review articles when it would be more appropriate to cite original papers. Not only is it important to credit the original work but citing a review frustrates any reader who wants to access the original papers. Please check all references carefully.
- Please make it explicit when it is you that is proposing a model, hypothesis, or claim based on your synthesis of the literature, as opposed to ideas that have already been proposed or under investigation by others. For example, in lines 141, 174, 219, 280 – are these your conclusions?
- In several instances where data are summarized without any discussion. For example, lines 283-288, outlines findings related to PARP1, however, there’s no discussion of what these data mean in relation to PGC1-alpha. It may be obvious, but not to this reader. Another example occurs from lines 257-264.
- The beginning of Section 6 (lines 344-353) just repeats what has already been said previously, this could probably be deleted.
- In Section 6, Line 355, you mention the involvement of SIRT1-3 and SIRT6 in diabetes. However, only the role of SIRT1 is elaborated on in the remainder of the section. What about the other SIRTs?
- Section 7. If there are published trials or studies testing the effects of NAD+ supplements it might be useful to discuss them here in relation to your SIRT-PGC1-alpha model.
- What is an NAD+ barrel? (line 204). Do you mean sink?
Author Response
We thank the reviewer for their comments and accordingly revised the manuscript to address these concerns.
I have explained point by point the details of the revisions in the manuscript and my responses to your comments. R; reviewer comment, A; answer.
R1. In many instances throughout the text the authors reference other Review articles when it would be more appropriate to cite original papers. Not only is it important to credit the original work but citing a review frustrates any reader who wants to access the original papers. Please check all references carefully.
A1. We thank the reviewer's suggestion, as reviewer comments, we checked all references. We removed some references and cite new ones. Or new references have added in their cited sentences. However, some reference, especially in the introduction, did not change to original papers since there are generally accepted one.
R2. Please make it explicit when it is you that is proposing a model, hypothesis, or claim based on your synthesis of the literature, as opposed to ideas that have already been proposed or under investigation by others. For example, in lines 141, 174, 219, 280 – are these your conclusions?
R2-1. Line 145, as reviewer suggestion, we revised the sentence as “We suggest that PGC-1a1 is likely to trigger DNS….”
R2-2. Line 180, we revised the sentence as “Thus, we speculate that PGC-1α regulate not only NAD+ consumption via SIRT1 but also PARP and NAD+ production via NAMPT and NAMPT.”
R2-3. Line 219, we revised the sentence as “Overall, these studies lead us to speculate that MAS is linked to mitochondrial and metabolic functions.”
R2-4. Line 291, we revised the sentence as “we suggest that PGC-1α could facilitate a cellular condition that prevents unnecessary NAD+ consumption by PARP and saves cellular NAD+ level (Fig. 1).”
R3. In several instances where data are summarized without any discussion. For example, lines 283-288, outlines findings related to PARP1, however, there’s no discussion of what these data mean in relation to PGC1-alpha. It may be obvious, but not to this reader. Another example occurs from lines 257-264.
A3. We agreed with your comment. For lines 257-264, the second phrase was moved to before the first phrase. We also revised the sentence that “We suggest the hypothesis that…” and we added the sentence as “thus, further studies are required to clear this.”
R4. For lines 283-288, The beginning of Section 6 (lines 344-353) just repeats what has already been said
previously, this could probably be deleted.
A4. As the reviewer suggested the phrase was removed.
R5. In Section 6, Line 355, you mention the involvement of SIRT1-3 and SIRT6 in diabetes. However, only the role of SIRT1 is elaborated on in the remainder of the section. What about the other SIRTs?
A5. We removed section 6’s content and rewrite them.
R6. Section 7. If there are published trials or studies testing the effects of NAD+ supplements it might be useful to discuss them here in relation to your SIRT-PGC1-alpha model.
A6. We tried to find clinical studies that performed NAD+ supplements with exercise training. We only found three-animal study but that study was for exercise performance, not metabolic diseases. Thus, we did not add them to the text.
R7. What is an NAD+ barrel? (line 204). Do you mean sink?
A7. We changed the word to “warehouse”
Round 2
Reviewer 2 Report
The manuscript has been suitably revised.